# Identification of a Novel Class of Anti-Melanogenic Compounds, (*Z*)-5-(Substituted benzylidene)-3-phenyl-2-thioxothiazolidin-4-one Derivatives, and Their Reactive Oxygen Species Scavenging Activities

**DOI:** 10.3390/antiox11050948

**Published:** 2022-05-11

**Authors:** Yeongmu Jeong, Sojeong Hong, Hee Jin Jung, Sultan Ullah, YeJi Hwang, Heejeong Choi, Jeongin Ko, Jieun Lee, Pusoon Chun, Hae Young Chung, Hyung Ryong Moon

**Affiliations:** 1Laboratory of Medicinal Chemistry, Department of Manufacturing Pharmacy, College of Pharmacy, Pusan National University, Busan 46241, Korea; dassabn@pusan.ac.kr (Y.J.); wwjd0912@pusan.ac.kr (S.H.); yjw4238@pusan.ac.kr (Y.H.); heejeong@pusan.ac.kr (H.C.); jungin8633@pusan.ac.kr (J.K.); yijiun@pusan.ac.kr (J.L.); 2Department of Pharmacy, College of Pharmacy, Pusan National University, Busan 46241, Korea; hjjung2046@pusan.ac.kr (H.J.J.); hyjung@pusan.ac.kr (H.Y.C.); 3Department of Molecular Medicine, The Scripps Research Institute, Jupiter, FL 33458, USA; sullah@scripps.edu; 4College of Pharmacy and Inje Institute of Pharmaceutical Sciences and Research, Inje University, Gimhae 50834, Korea; pusoon@inje.ac.kr

**Keywords:** tyrosinase, ROS, radical scavenging, docking simulation, anti-melanogenesis, dual mechanism, kojic acid

## Abstract

The rate-determining role of tyrosinase makes it a critical component in the mechanism that is responsible for melanogenesis. Thirteen (*Z*)-5-(substituted benzylidene)-3-phenyl-2-thioxothiazolidin-4-one ((*Z*)-BPTT) analogs were designed based on the structural features of two potent tyrosinase inhibitors, viz. (Z)-5-(3-hydroxy-4-methoxybenzylidene)-2-thioxothiazolidin-4-one (5-HMT) and (*Z*)-2-(2,4-dihydroxybenzylidene)benzo[4,5]imidazo[2,1-*b*]thiazol-3(2*H*)-one (compound I). The trisubstituted double bond geometry of the (*Z*)-BPTT analogs that were generated by Knoevenagel condensation was determined using vicinal ^1^H and ^13^C coupling constants in ^13^C NMR spectra. Four analogs, numbers **1**–**3** and **6,** inhibited mushroom tyrosinase 9 to 29 times more potently than kojic acid did. Kinetic study results indicated that these four analogs inhibited mushroom tyrosinase competitively and this was supported by docking simulation. Also, docking results using human tyrosinase suggested that analogs **2** and **3** might be potent human tyrosinase inhibitors. In vitro studies using B16F10 cells (a melanoma cell line) showed that analogs **1**, **2**, **3**, and **6** inhibited cellular tyrosinase and melanin production more than kojic acid did, without perceptible cytotoxicity. In particular, analog **2**, which possesses a catechol group, exerted an extremely potent anti-melanogenic effect. In addition, analog **2** showed strong scavenging activity against DPPH and ABTS radicals. Furthermore, analog **2** not only reduced ROS levels, which induce melanogenesis, but it also suppressed tyrosinase and MITF (microphthalamia-associated transcription factor) protein levels and the expressions of melanogenesis-related genes. These results suggest that analog **2** is an efficient tyrosinase inhibitor that alleviates melanogenesis by dual mechanisms of (i) the inhibition of melanogenesis-related proteins and genes and (ii) the direct inhibition of tyrosinase activity.

## 1. Introduction

Melanin is a dark brown pigment that is predominantly responsible for skin color and is also found in hair, eyes, ears, and the brain [1,2]. Melanin is produced by the conversion of the amino acid l-tyrosine to l-DOPA (3,4-dihydroxyphenylalanine) and its oxidation to dopaquinone [3,4], the precursor of melanin formation. Tyrosinase is a multifunctional copper-containing enzyme that catalyzes the rate-limiting step of melanin biosynthesis; that is, the conversion of l-tyrosine to melanin. In humans, tyrosinase overexpression leads to the accumulation of melanin in skin and can trigger hyperpigmentation manifesting as freckles, melasma, age spots, postinflammatory hyperpigmentation, or melanoma [5,6]. MITF (microphthalmia-associated transcription factor) is a master regulator of melanocyte survival and proliferation and plays a key role in melanogenesis as a regulator of tyrosinase and the tyrosinase-related proteins (TRP)-1 and TRP-2 [7,8,9,10]. Therefore, research efforts have focused on identifying agents that regulate the levels of these melanogenic proteins and tyrosinase activity [11,12].

Ultra-violet (UV) radiation generates hydrogen peroxide (H_2_O_2_) and hydroxyl (HO^•^) and superoxide (O_2_^•−^) radicals in melanocytes and, thus, initiates melanin biosynthesis, which primarily results in the production of pheomelanin [13,14]. Free radicals also participate in the biosynthesis of melanin and are involved in the catalytic conversion of l-DOPA to dopaquinone by tyrosinase. Thus, increasing the free radical concentrations or activities in living organisms can increase melanin production [15,16,17]. On the other hand, the tyrosinase-inhibiting and free radical scavenging effects of antioxidants inhibit melanin production.

Researchers have devised many strategies to inhibit melanogenesis, such as inhibiting melanosome maturation, melanosome transfer from melanocytes to keratinocytes, and tyrosinase expression or activity [18]. However, interventions that target melanosome maturation and melanosome transfer have unwanted side effects such as burning, swelling, unusual discoloration, itching, and redness [19] and, therefore, research has been primarily directed toward identifying suitable tyrosinase inhibitors [20,21,22,23,24,25,26,27,28]. As a result, several tyrosinase inhibitors (natural, semi-synthetic, and synthetic) have been identified. However, all of these have side effects and, thus, only a handful with moderate side effects are used as whitening agents [29,30]. Therefore, more potent novel tyrosinase inhibitors with better safety profiles are needed.

As shown in Figure 1, 5-HMT ((*Z*)-5-(3-hydroxy-4-methoxybenzylidene)-2-thioxothiazolidin-4-one), which possesses a β-phenyl-α,β-unsaturated carbonyl (PUSC) scaffold [20,23], was reported to have a lower IC_50_ value (18.1 ± 1.0 μM) against mushroom tyrosinase than kojic acid, a well-known natural tyrosinase inhibitor, and it had a skin-lighting effect in UVB-irradiated HRM2 hairless mice [31]. Compound **I**, ((*Z*)-2-(2,4-dihydroxybenzylidene)benzo[4,5]imidazo[2,1-*b*]thiazol-3(2*H*)-one), also had a low IC_50_ value (5.0 ± 0.38 μM) against mushroom tyrosinase and exhibited anti-tyrosinase and anti-melanogenic effects in B16F10 murine melanoma cells [32]. Based on the structural characteristics of these two compounds, (*Z*)-5-(substituted benzylidene)-3-phenyl-2-thioxothiazolidin-4-one ((*Z*)-BPTT) analogs were designed as a new class of tyrosinase inhibitors. Compound **I** had a structure that is similar to that of 5-HMT and the cleavage of the N-C bond of compound **I** (indicated by a broken green line in Figure 1) afforded the (*Z*)-BPTT analogs similar structures.

Thus, we designed and synthesized 13 (*Z*)-BPTT analogs using the structural features of 5-HMT and compound **I**, with different substituents on the benzylidene group**,** determined their IC_50_ values against mushroom tyrosinase, and evaluated their tyrosinase activities and melanogenesis inhibition effects in B16F10 cells. Mechanisms were also investigated using kinetic studies and docking simulations using mushroom tyrosinase and a homologous human tyrosinase model. Furthermore, we examined the inhibitory effects of these analogs on cellular tyrosinase activity and melanin production and their abilities to scavenge ROS and radicals such as DPPH and ABTS. Finally, the effects of the analogs on the levels of proteins and the expressions of genes that are related to melanogenesis in B16F10 cells were examined.

## 2. Materials and Methods

### 2.1. General Methods

All of the reagents that were used were purchased from Sigma-Aldrich (St. Louis, MO, USA), Daejung Chemical & Materials Co. Ltd. (Gyeonggi-do, Korea), SEJIN CI Co. (Seoul, Korea), or ThermoFisher Scientific (Seoul, Korea) and used without further purification. The anhydrous solvents, including dichloromethane, were distilled over CaH_2_ or Na/benzophenone prior to their use. The reactions were performed under argon or nitrogen and monitored by thin layer chromatography (TLC, Merck pre-coated 60F_245_ plates). Flash column chromatography was performed using silica gel (MP Silica 40–63, 60 Å). High resolution mass spectroscopy (HRMS) data were obtained using an Agilent Accurate Mass quadruple time of flight (Q-TOF) mass spectrometer that was equipped with a liquid chromatograph (Agilent, Santa Clara, CA, USA) in electrospray ionization positive (ESI+) mode. Low-resolution mass spectroscopy (LRMS) data were obtained in electrospray ionization (ESI) positive or negative mode using an Expression CMS mass spectrometer (Advion Ithaca, NY, USA). The ^1^H and ^13^C NMR spectral data were obtained using a Varian Unity AS500 unit (Agilent technologies, Santa Clara, CA, USA) at 125 MHz for ^13^C NMR and 500 MHz for ^1^H NMR. CDCl_3_ and DMSO-*d*_6_ were used as NMR solvents. All of the chemical shifts are presented in parts per million (ppm) versus residual solvent or deuterated peaks (*δ*_H_ 7.27 and *δ*_C_ 77.0 for CDCl_3_, and *δ*_H_ 2.48 and *δ*_C_ 39.9 for DMSO-*d*_6_). Coupling constants (*J*) are expressed in hertz (Hz). The following abbreviations are used for ^1^H NMR: s (singlet), brs (broad singlet), d (doublet), t (triplet), q (quartet), dd (doublet of doublets), and m (multiplet).

#### 2.1.1. Synthesis of Compound **14**

To a stirred solution of phenyl isothiocyanate (5.0 g, 36.98 mmol) and thioglycolic acid (2.6 mL, 36.97 mmol) in dichloromethane (100 mL), Et_3_N (5.2 mL, 36.95 mmol) was added at 0 ^o^C. The reaction mixture was stirred at ambient temperature for 2 d and partitioned between dichloromethane and water. The organic layer was then dried over MgSO_4_, filtered, and concentrated under reduced pressure. The solid was recrystallized from hexane and dichloromethane in order to afford compound **14** (3.197 g, 41%) as a solid.

#### 2.1.2. General Synthetic Procedure for Analogs **1**–**2** and **4**–**13**

An acetic acid (1.5 mL) solution of **14** (90 mg, 0.43 mmol) containing an appropriate benzaldehyde (1.0 equiv.) was refluxed in the presence of NaOAc (3.0 equiv.) for 1–7 h. After cooling, water was added to the reaction mixture and the resultant solid was filtered and washed with water so as to afford compounds **1**–**2** and **4**–**13** as solids.

#### 2.1.3. Synthesis of Analog **3**

An ethanoic solution (2 mL) of **14** (90 mg, 0.43 mmol) and 2,4-dihydroxybenzaldehyde (65 mg, 0.47 mmol) in EtOH (2 mL) was refluxed in the presence of piperidine (0.01 mL, 0.10 mmol), as a catalyst, for 4 h. After cooling, water was added to the reaction mixture and the resultant solid was filtered and washed with water and dichloromethane in order to afford compound **3** (42 mg, 30%) as a solid.

#### 2.1.4. Characterization of Analogs **1**–**14**

##### (Z)-5-(4-Hydroxybenzylidene)-3-phenyl-2-thioxothiazolidin-4-one (**1**)

Reaction time: 5 h; ^1^H NMR (500 MHz, DMSO-*d*_6_) *δ* 10.49 (s, 1H, OH), 7.75 (s, 1H, vinylic H), 7.57–7.54 (m, 4H, 3-H, 5-H, 2′-H, 6′-H), 7.51 (t, 1H, *J* = 7.5 Hz, 4-H), 7.40 (d, 2H, *J* = 7.5 Hz, 2-H, 6-H), 6.97 (d, 2H, *J* = 9.0 Hz, 3′-H, 5′-H); ^13^C NMR (125 MHz, DMSO-*d*_6_) *δ* 194.3, 167.5, 161.1, 135.8, 134.0, 133.8, 129.8, 129.7, 129.2, 124.6, 119.0, 117.1; LRMS (ESI–) *m*/*z* 312 (M–H)^–^; HRMS (ESI+) *m*/*z* C_16_H_11_NO_2_S_2_ (M + H)^+^ calcd 314.0304, obsd 314.0307.

##### (Z)-5-(3,4-Dihydroxybenzylidene)-3-phenyl-2-thioxothiazolidin-4-one (**2**)

Reaction time: 6 h; ^1^H NMR (500 MHz, DMSO-*d*_6_) *δ* 9.81 (brs, 2H, 2×OH), 7.66 (s, 1H, vinylic H), 7.58–7.48 (m, 3H, 3-H, 4-H, 5-H), 7.39 (d, 2H, *J* = 7.5 Hz, 2-H, 6-H), 7.11–7.07 (m, 2H, 2′-H, 6′-H), 6.92 (d, 1H, *J* = 7.5 Hz, 5′-H); ^13^C NMR (125 MHz, DMSO-*d*_6_) *δ* 194.3, 167.5, 150.0, 146.6, 135.8, 134.4, 129.8, 129.7, 129.2, 125.8, 125.0, 118.7, 117.1, 117.0; LRMS (ESI–) *m*/*z* 328 (M–H)^–^; HRMS (ESI+) *m*/*z* C_16_H_11_NO_3_S_2_ (M + H)^+^ calcd 330.0253, obsd 330.0248.

##### (Z)-5-(2,4-Dihydroxybenzylidene)-3-phenyl-2-thioxothiazolidin-4-one (**3**)

Reaction time: 4 h; ^1^H NMR (500 MHz, DMSO-*d*_6_) *δ* 10.71 (brs, 1H, OH), 10.40 (brs, 1H, OH), 7.97 (s, 1H, vinylic H), 7.59–7.47 (m, 3H, 3-H, 4-H, 5-H), 7.37 (d, 2H, *J* = 7.5 Hz, 2-H, 6-H), 7.26 (d, 1H, *J* = 8.0 Hz, 6′-H), 6.49–6.44 (m, 2H, 3′-H, 5′-H); ^13^C NMR (125 MHz, DMSO-*d*_6_) *δ* 194.5, 167.7, 163.1, 160.4, 136.0, 132.0, 129.7, 129.7, 129.2, 129.2, 116.8, 112.5, 109.5, 103.0; LRMS (ESI–) *m*/*z* 328 (M–H)^–^.

##### (Z)-5-(4-Hydroxy-3-methoxybenzylidene)-3-phenyl-2-thioxothiazolidin-4-one (**4**)

Reaction time: 3 h; ^1^H NMR (500 MHz, DMSO-*d*_6_) *δ* 10.16 (s, 1H, OH), 7.76 (s, 1H, vinylic H), 7.56 (t, 2H, *J* = 7.5 Hz, 3-H, 5-H), 7.51 (t, 1H, *J* = 7.5 Hz, 4-H), 7.40 (d, 2H, *J* = 7.5 Hz, 2-H, 6-H), 7.25 (s, 1H, 2′-H), 7.17 (d, 1H, *J* = 8.0 Hz, 6′-H), 6.98 (d, 1H, *J* = 8.0 Hz, 5′-H), 3.85 (s, 3H, OCH_3_); ^13^C NMR (125 MHz, DMSO-*d*_6_) *δ* 194.2, 167.5, 150.7, 148.6, 135.8, 134.3, 129.8, 129.8, 129.2, 125.6, 125.0, 119.2, 116.9, 115.2, 56.1; LRMS (ESI–) *m*/*z* 342 (M–H)^–^; HRMS (ESI+) *m*/*z* C_17_H_13_NO_3_S_2_ (M + H)^+^ calcd 344.0410, obsd 344.0408.

##### (Z)-5-(3-Ethoxy-4-hydroxybenzylidene)-3-phenyl-2-thioxothiazolidin-4-one (**5**)

Reaction time: 5 h; ^1^H NMR (500 MHz, DMSO-*d*_6_) *δ* 10.07 (s, 1H, OH), 7.74 (s, 1H, vinylic H), 7.54 (t, 2H, *J* = 7.5 Hz, 3-H, 5-H), 7.50 (t, 1H, *J* = 7.5 Hz, 4-H), 7.38 (d, 2H, *J* = 7.5 Hz, 2-H, 6-H), 7.22 (d, 1H, *J* = 2.0 Hz, 2′-H), 7.16 (dd, 1H, *J* = 8.5, 2.0 Hz, 6′-H), 6.98 (d, 1H, *J* = 8.5 Hz, 5′-H), 4.09 (q, 2H, *J* = 7.0 Hz, OC*H_2_*CH_3_), 1.37 (t, 3H, *J* = 7.0 Hz, OCH_2_C*H_3_*); ^13^C NMR (125 MHz, DMSO-*d*_6_) *δ* 192.3, 166.7, 147.7, 145.4, 134.0, 133.1, 128.6, 128.5, 127.4, 125.3, 124.9, 118.9, 114.3, 111.8, 63.8, 13.7; LRMS (ESI–) *m*/*z* 356 (M–H)^–^; HRMS (ESI+) *m*/*z* C_18_H_15_NO_3_S_2_ (M + H)^+^ calcd 358.0566, obsd 358.0557.

##### (Z)-5-(3-Hydroxy-4-methoxybenzylidene)-3-phenyl-2-thioxothiazolidin-4-one (**6**)

Reaction time: 6 h; ^1^H NMR (500 MHz, DMSO-*d*_6_) *δ* 9.59 (s, 1H, OH), 7.67 (s, 1H, vinylic H), 7.53 (t, 2H, *J* = 7.5 Hz, 3-H, 5-H), 7.48 (t, 1H, *J* = 7.5 Hz, 4-H), 7.38 (d, 2H, *J* = 7.5 Hz, 2-H, 6-H), 7.17 (dd, 1H, *J* = 8.5, 2.0 Hz, 6′-H), 7.10 (m, 2H, 2′-H, 5′-H), 3.83 (s, 3H, OCH_3_); ^13^C NMR (125 MHz, DMSO-*d*_6_) *δ* 194.3, 167.4, 151.2, 147.6, 135.8, 133.9, 129.8, 129.7, 129.2, 126.2, 125.2, 120.1, 116.5, 113.1, 56.2; LRMS (ESI–) *m*/*z* 342 (M–H)^–^; HRMS (ESI+) *m*/*z* C_17_H_13_NO_3_S_2_ (M + H)^+^ calcd 344.0410, obsd 344.0408.

##### (Z)-5-(4-Methoxybenzylidene)-3-phenyl-2-thioxothiazolidin-4-one (**7**)

Reaction time: 5 h; ^1^H NMR (500 MHz, DMSO-*d*_6_) *δ* 7.80 (s, 1H, vinylic H), 7.65 (d, 2H, *J* = 9.0 Hz, 2′-H, 6′-H), 7.54 (t, 2H, *J* = 7.5 Hz, 3-H, 5-H), 7.50 (t, 1H, *J* = 7.5 Hz, 4-H), 7.39 (d, 2H, *J* = 7.5 Hz, 2-H, 6-H), 7.14 (d, 2H, *J* = 9.0 Hz, 3′-H, 5′-H), 3.84 (s, 3H, OCH_3_); ^13^C NMR (125 MHz, CDCl_3_) *δ* 192.5, 166.7, 160.8, 134.0, 132.5, 131.8, 128.6, 128.5, 127.4, 125.0, 119.2, 114.0, 54.5; LRMS (ESI+) *m*/*z* 328 (M + H)^+^; HRMS (ESI+) *m*/*z* C_17_H_13_NO_2_S_2_ (M + H)^+^ calcd 328.0460, obsd 328.0457.

##### (Z)-5-(3,4-Dimethoxybenzylidene)-3-phenyl-2-thioxothiazolidin-4-one (**8**)

Reaction time: 7 h; ^1^H NMR (500 MHz, DMSO-*d*_6_) *δ* 7.79 (s, 1H, vinylic H), 7.54 (t, 2H, *J* = 7.5 Hz, 3-H, 5-H), 7.49 (t, 1H, *J* = 7.5 Hz, 4-H), 7.39 (d, 2H, *J* = 7.5 Hz, 2-H, 6-H), 7.29–7.25 (m, 2H, 2′-H, 6′-H), 7.16 (d, 1H, *J* = 8.0 Hz, 5′-H), 3.84 (s, 3H, OCH_3_), 3.82 (s, 3H, OCH_3_); ^13^C NMR (125 MHz, CDCl_3_) *δ* 193.3, 167.7, 151.7, 149.6, 135.0, 133.8, 129.7, 129.6, 128.4, 126.3, 125.6, 120.4, 112.5, 111.5, 56.1, 56.0; LRMS (ESI+) *m*/*z* 358 (M + H)^+^; HRMS (ESI+) *m*/*z* C_18_H_15_NO_3_S_2_ (M + H)^+^ calcd 358.0566, obsd 358.0566.

##### (Z)-5-(2,4-Dimethoxybenzylidene)-3-phenyl-2-thioxothiazolidin-4-one (**9**)

Reaction time: 7 h; ^1^H NMR (500 MHz, CDCl_3_) *δ* 8.11 (s, 1H, vinylic H), 7.55 (t, 2H, *J* = 7.5 Hz, 3-H, 5-H), 7.50 (t, 1H, *J* = 7.5 Hz, 4-H), 7.41 (d, 1H, *J* = 8.5 Hz, 6′-H), 7.29 (d, 2H, *J* = 7.5 Hz, 2-H, 6-H), 6.62 (dd, 1H, *J* = 8.5, 2.0 Hz, 5′-H), 6.49 (d, 1H, *J* = 2.0 Hz, 3′-H), 3.92 (s, 3H, OCH_3_), 3.89 (s, 3H, OCH_3_); ^13^C NMR (125 MHz, DMSO-*d*_6_) *δ* 194.7, 167.6, 164.3, 160.5, 135.9, 132.3, 129.8, 129.7, 129.2, 128.7, 119.9, 114.9, 107.6, 99.2, 56.4, 56.3; LRMS (ESI+) *m*/*z* 358 (M + H)^+^; HRMS (ESI+) *m*/*z* C_18_H_15_NO_3_S_2_ (M + H)^+^ calcd 358.0566, obsd 358.0570.

##### (Z)-3-Phenyl-2-thioxo-5-(3,4,5-trimethoxybenzylidene)thiazolidin-4-one (**10**)

Reaction time: 5 h; ^1^H NMR (500 MHz, DMSO-*d*_6_) *δ* 7.78 (s, 1H, vinylic H), 7.54 (t, 2H, *J* = 7.5 Hz, 3-H, 5-H), 7.50 (t, 1H, *J* = 7.5 Hz, 4-H), 7.39 (d, 2H, *J* = 7.5 Hz, 2-H, 6-H), 6.98 (s, 2H, 2′-H, 6′-H), 3.85 (s, 6H, 2×OCH_3_), 3.74 (s, 3H, OCH_3_); ^13^C NMR (125 MHz, DMSO-*d*_6_) *δ* 194.2, 167.4, 153.8, 140.4, 135.7, 133.5, 130.0, 129.8, 129.2, 129.0, 122.6, 108.6, 60.7, 56.6; LRMS (ESI+) *m*/*z* 388 (M + H)^+^; HRMS (ESI+) *m*/*z* C_19_H_17_NO_4_S_2_ (M + H)^+^ calcd 388.0672, obsd 388.0682.

##### (Z)-5-(4-Hydroxy-3,5-dimethoxybenzylidene)-3-phenyl-2-thioxothiazolidin-4-one (**11**)

Reaction time: 5 h; ^1^H NMR (500 MHz, DMSO-*d*_6_) *δ* 9.56 (s, 1H, OH), 7.74 (s, 1H, vinylic H), 7.55 (t, 2H, *J* = 7.5 Hz, 3-H, 5-H), 7.50 (t, 1H, *J* = 7.5 Hz, 4-H), 7.39 (d, 2H, *J* = 7.5 Hz, 2-H, 6-H), 6.94 (s, 2H, 2′-H, 6′-H), 3.84 (s, 6H, 2 × OCH_3_); ^13^C NMR (125 MHz, CDCl_3_) *δ* 192.2, 166.6, 146.5, 136.9, 133.9, 133.1, 128.7, 128.6, 127.3, 123.9, 119.4, 106.9, 55.4; LRMS (ESI–) *m*/*z* 372 (M–H)^–^; HRMS (ESI+) *m*/*z* C_18_H_15_NO_4_S_2_ (M + H)^+^ calcd 374.0515, obsd 374.0508.

##### (Z)-5-(3-Bromo-4-hydroxybenzylidene)-3-phenyl-2-thioxothiazolidin-4-one (**12**)

Reaction time: 4 h; ^1^H NMR (500 MHz, DMSO-*d*_6_) *δ* 11.30 (s, 1H, OH), 7.86 (s, 1H, 2′-H), 7.72 (s, 1H, vinylic H), 7.55–7.47 (m, 4H, 3-H, 4-H, 5-H, 6′-H), 7.37 (d, 2H, *J* = 7.5 Hz, 2-H, 6-H), 7.12 (d, 1H, *J* = 9.0 Hz, 5′-H); ^13^C NMR (125 MHz, DMSO-*d*_6_) *δ* 194.0, 167.4, 157.3, 136.8, 135.7, 132.3, 131.5, 129.9, 129.8, 129.2, 126.2, 120.8, 117.7, 110.9; LRMS (ESI–) *m*/*z* 390 (M–H)^–^, 392 (M + 2–H)^–^; HRMS (ESI+) *m*/*z* C_16_H_10_BrNO_2_S_2_ (M + H)^+^ calcd 391.9409, obsd 391.9405, (M + 2+H)^+^ calcd 393.9388, obsd 393.9361.

##### (Z)-5-(3,5-Dibromo-4-hydroxybenzylidene)-3-phenyl-2-thioxothiazolidin-4-one (**13**)

Reaction time: 1 h; ^1^H NMR (500 MHz, DMSO-*d*_6_) *δ* 10.93 (brs, 1H, OH), 7.84 (s, 2H, 2′-H, 6′-H), 7.73 (s, 1H, vinylic H), 7.54 (t, 2H, *J* = 7.5 Hz, 3-H, 5-H), 7.51 (t, 1H, *J* = 7.5 Hz, 4-H), 7.38 (d, 2H, *J* = 7.5 Hz, 2-H, 6-H); ^13^C NMR (125 MHz, DMSO-*d*_6_) *δ* 193.6, 167.2, 153.7, 135.6, 134.8, 130.5, 129.9, 129.8, 129.2, 127.8, 122.8, 113.0; LRMS (ESI–) *m*/*z* 468 (M–H)^–^, 470 (M + 2–H)^–^, 472 (M + 4–H)^–^.

##### 3-Phenyl-2-thioxothiazolidin-4-one (**14**)

^1^H NMR (500 MHz, DMSO-*d*_6_) *δ* 7.51 (t, 2H, *J* = 7.5 Hz, 3-H, 5-H), 7.46 (t, 1H, *J* = 7.5 Hz, 4-H), 7.26 (d, 2H, *J* = 7.5 Hz, 2-H, 6-H), 4.37 (s, 2H, CH_2_); ^13^C NMR (125 MHz, DMSO-*d*_6_) *δ* 204.1, 174.5, 136.0, 129.7, 129.7, 129.2, 37.6; LRMS (ESI–) *m*/*z* 208 (M–H)^–^.

### 2.2. Biological Evaluation and Docking Simulation

#### 2.2.1. Reagents

The kojic acid, trolox, l-ascorbic acid, l-4-hydroxyphenylalanine (l-tyrosine), l-3,4-dihydroxyphenylalanine (l-DOPA), alpha-melanocyte stimulating hormone (α-MSH), 3-isobutyl-1-methylxanthine (IBMX), 4-(1,1,3,3-tetramethylbutyl)phenylpolyethylene glycol (Triton™ X-100), phenylmethylsulfonyl fluoride (PMSF), 3-morpholinosydnonimine (SIN-1), dimethyl sulfoxide (DMSO), potassium hydrogen phosphate, potassium dihydrogen phosphate, 2,2′-azino-bis(3-ethylbenzothiazoline-6-sulfonic acid) (ABTS), and mushroom tyrosinase were all purchased from Sigma-Aldrich (St. Louis, MO, USA).

#### 2.2.2. Mushroom Tyrosinase Inhibition Assay

In order to examine the inhibitory effect of the (*Z*)-BPTT analogs on tyrosinase activity, commercially available mushroom tyrosinase (Sigma-Aldrich, St. Louis, MO, USA) was used. The mushroom tyrosinase-inhibitory activities of the analogs were determined as described previously [33] with modification. Briefly, 20 µL (20 units) of an aqueous mushroom tyrosinase solution was added to each well of a 96-well microplate containing 170 µL of a substrate mixture consisting of 345 μM of l-tyrosine solution, 17.2 mM phosphate buffer (pH 6.5), and 10 µL of a (*Z*)-BPTT analog solution at different concentrations or 10 µL of a kojic acid solution (2, 10, or 50 μM). After incubating the assay mixtures for 30 min at 37 °C, the remaining tyrosinase activities were calculated from the amounts of dopachrome that were formed as determined by measuring their absorbance at 492 nm using a microplate reader (VersaMax™, Molecular Devices, Sunnyvale, CA, USA). The curves of the IC_50_ values that were derived from the X-axis of the inhibitor concentration versus the amount of product that was produced were determined by aligning the dose–response curve with the Y-axis. Inhibition experiments were performed in triplicate using three to six different concentrations of the (*Z*)-BPTT analogs or kojic acid. Log–linear curves and their equations were determined by using inhibition percentages at three to six analog concentrations. The individual IC_50_ values were determined by reading the concentrations on the Y-axis at 50% inhibition. The results are the means of three independent experiments.

#### 2.2.3. Kinetic Studies of Mushroom Tyrosinase Inhibition by Analogs **1**–**3** and **6**

The Michael constants (K_M_), inhibition constants (K_i_), and Lineweaver–Burk and Dixon plots of the analogs were used in order to determine their modes of action [34,35]. In brief, 10 µL of the test analogs **1**–**3** and **6** (at final concentrations of 0, 1, 2, or 4 µM for analogs **1** and **6**; 0, 2.5, 5, or 10 µM for analog **2;** and 0, 0.4, 0.8, or 1.6 µM for analog **3**) was added to a 96-well plate containing a substrate mixture (170 µL) containing an aqueous solution of l-DOPA at final concentrations of 0.5, 1, 2, 4, 8, or 16 mM, along with 14.7 mM of potassium phosphate buffer (pH 6.5) and mushroom tyrosinase solution (20 µL, 20 units). The initial rates of dopachrome production in the reaction mixtures were calculated by measuring the increases in absorbance at 492 nm (ΔOD_492_/min) using a microplate reader (VersaMax™, Molecular Devices, Sunnyvale, CA, USA). The maximal reaction velocity (V_max_) was determined using Lineweaver–Burk plots (plots of the inverse of the reaction rate (1/V) versus the inverse of the substrate concentration (1/[S])) that were obtained using six different l-DOPA concentrations. The mode of action of each analog was determined using the convergence point of the four plot lines.

#### 2.2.4. In Silico Study of the Interactions between Mushroom Tyrosinase and Analogs **1**–**3**, Analog **6,** or Kojic Acid

The in silico study was performed using Schrödinger Suite (2021-1) as previously described [36] with slight modification. The crystal structure of mushroom tyrosinase (*m*TYR) (*Agaricus bisporus*, PDB: 2Y9X) was downloaded from the Protein Data Bank (PDB) database into Maestro 12.4 Protein Preparation Wizard. The crystal structure of *m*TYR was processed and the unwanted protein chains were removed. In order to optimize the structure, hydrogen atoms were added and water molecules more than 3 Å away from the enzyme were removed. The Glide grid and the active site of the enzyme was determined using the binding site of the ligand (tropolone) as described by the PDB and literature [37,38]. The structures of analogs **1**–**3** and kojic acid were imported in CDXML format from Maestro and prepared using LigPrep before ligand docking. The analogs and kojic acid were then docked to the Glide grid of *m*TYR using Glide from the task list [37]. The ligand–protein interactions and binding affinities were obtained using the Glide extra precision (XP) method [39].

#### 2.2.5. In Silico Study of Interactions between the Human Tyrosinase Homology Model and Analogs **1**–**3**, Analog **6,** and Kojic Acid

The *h*TYR homology model was generated using the Swiss-Model online server and Schrödinger Suite (2020-2). The protein sequence of *h*TYR (P14679) was obtained from the UniProt database and the homology model was generated on the Swiss-Model online server and based on TRP1 as a template (PDB ID: 5M8Q). The homology model was further processed using Schrödinger Suite and validated using Schrödinger prime (the homology modeling tool in Schrödinger Suite). Analogs **1**–**3** and kojic acid were docked with the newly generated human homology model using the same docking protocols that were mentioned above for *m*TYR.

#### 2.2.6. Cell Culture

B16F10 cells (a murine melanoma cell line) were purchased from the American Type Culture Collection (ATCC, Manassas, VA, USA). Dulbecco’s modified Eagle’s medium (DMEM), fetal bovine serum (FBS), phosphate buffer solution (PBS), trypsin, penicillin, and streptomycin were obtained from Gibco (Grand Island, NY, USA). The B16F10 cells were cultured in DMEM containing streptomycin/penicillin (100 µg/mL/100 IU/mL) and 10% heat-inactivated FBS in a humidified 5% CO_2_ atmosphere at 37 °C. Cell viability, anti-melanogenesis activity, and anti-tyrosinase activity assays were performed on cells in 6- or 96-well culture plates.

#### 2.2.7. Cell Viability Assay

Cell viability assays on the (*Z*)-BPTT analogs **1**–**3** and **6** were performed on B16F10 melanoma cells using the EZ-Cytox assay (DoGenBio, Seoul, Korea), as previously described [40]. Briefly, B16F10 cells were seeded in a 96-well plate and cultured for 24 h in a humidified 5% CO_2_ atmosphere at 37 °C. On the following day, the cells were exposed to six concentrations (0, 1, 2, 5, 10, or 20 µM) of analogs **1**–**3** or **6** and then incubated in a humidified 5% CO_2_ atmosphere for 24 h or 48 h at 37 °C. Then, 10 µL of the EZ-Cytox solution was added to each well and the cells were incubated for 2 h at 37 °C. The absorbances were measured at 450 nm using a microplate reader (VersaMax™, Molecular Devices, Sunnyvale, CA, USA) in order to calculate the cells’ viabilities. These assays were performed independently in triplicate.

#### 2.2.8. Melanin Content Assays

The effects of the (*Z*)-BPTT analogs **1**–**3** and **6** on cellular melanin production were determined using a standard protocol [41] with some modification. Briefly, B16F10 cells were seeded at a density of 1 × 10^5^ cells per well in 6-well plates and allowed to adhere to the well bases under the same conditions that were used for cell culturing. After culturing for 24 h, the cells were exposed to analogs **1**–**3** or analog **6** (at 0, 2.5, 5, 10, or 20 µM), or kojic acid (10 or 20 µM), for 1 h and co-stimulated with 1 µM of α-MSH and 200 µM of IBMX in a humidified 5% CO_2_ atmosphere for 48 h at 37 °C. In order to determine the melanin contents, the B16F10 cells that were exposed to α-MSH and IBMX for 48 h were rinsed with PBS twice and incubated in 200 µL of 1 N NaOH solution containing 10% dimethyl sulfoxide (DMSO) for 1 h at 60 °C. The cell lysates were transferred to a 96-well plate and the absorbances at 405 nm were measured using a microplate reader (VersaMax™, Molecular Devices, Sunnyvale, CA, USA). These experiments were performed individually in triplicate.

#### 2.2.9. Evaluation of Cellular Tyrosinase Activity

The cellular tyrosinase activity was examined by assessing the oxidation rate of L-DOPA, as was previously described [42], with a slight modification. In brief, B16F10 cells were seeded at a density of 1 × 10^5^ cells/well in a 6-well plate and allowed to adhere to the well bases using the same conditions that were used for cell culturing. After 24 h of incubation, the B16F10 cells were exposed to various concentrations (0, 5, 10, or 20 µM) of analogs **1**–**3**, and analog **6,** or kojic acid (20 µM). After incubation for 1 h, tyrosinase activities were stimulated by co-treating the cells with α-MSH (1 µM) and IBMX (200 µM). After incubation for 48 h under the same conditions that were used for cell culturing, the cells were rinsed twice with PBS and then exposed to 100 µL of a lysis buffer solution (90 µL of 50 mM phosphate buffer in pH 6.5, 5 µL of 2 mM PMSF, and 5 µL of 20% Triton X-100) and held for 30 min at −80 °C. After defrosting, the cell lysates were centrifuged at 12,000 rpm for 30 min at 4 °C. Then, 10 mM l-DOPA (20 µL) was mixed with the supernatants (80 µL) of the lysates in a 96-well plate. The absorbances of these mixtures were measured at 492 nm every 10 min for 1 h at 37 °C using a microplate reader (VersaMax™, Molecular Devices, Sunnyvale, CA, USA). All of these experiments were conducted individually in triplicate.

#### 2.2.10. DPPH Radical Scavenging Activity Assay

The effects of the (*Z*)-BPTT analogs **1**–**13** on DPPH radical scavenging activities were investigated as previously described with slight modification [43]. Briefly, 180 μL of 0.2 mM DPPH in methanol was added to 96-well plates and then 20 μL of one of the (*Z*)-BPTT analogs **1**–**13** (10 mM in DMSO) or the positive control (10 mM l-ascorbic acid in distilled water) was added and mixed. The 96-well plates were incubated in the dark for 30 min and then the well optical densities were measured at 517 nm using a VersaMax™ microplate reader (Molecular Devices, Sunnyvale, CA, USA). All of these experiments were performed independently in triplicate. The DPPH radical scavenging activities of the analogs were calculated using the following formula:Scavenging activity (%) = (Ac − As) × 100/Ac
where Ac is the optical density of the non-treated control and As is the optical density of the tested material.

#### 2.2.11. ABBS Free Radical Scavenging Activity Assay

ABTS free radical scavenging activity was used to evaluate the antioxidant activities of analogs **1**–**13**, as previously described [44]. To generate ABTS free radicals, 10 mL of 7 mM ABTS was mixed with 10 mL of 2.45 mM potassium persulfate in distilled water and incubated in the dark for 16 h at room temperature. This ABTS free radical solution was then diluted with absolute methanol to an absorbance of 0.70 ± 0.02 at 734 nm. Aliquots (10 µL) of the test samples (10% DMSO + 90% EtOH) were added at a final concentration of 100 µM to 90 µL of the diluted ABTS free radical solution and incubated for 2 min in the dark at room temperature. Absorbances were then measured at 734 nm using a VersaMax™ microplate reader (Molecular Devices, Sunnyvale, CA, USA). The ABTS free radical scavenging activities of the test analogs were calculated using the following formula:Scavenging activity (%) = (Ac − As) × 100/Ac
where Ac is the optical density of the non-treated control and As is the optical density of the tested material.

#### 2.2.12. ROS Scavenging Activity Assay

ROS scavenging activity was determined as has previously been described by Ali et al. [45] and Lebel and Bondy [46]. Briefly, DCFH-DA (2′,7′-dichlorodihydrofluorescein-diacetate) (2.5 mM; Molecular Probes, Eugene, OR, USA) mixed with esterase (1.5 units/mL) in a phosphate buffer (50 mM; pH 7.4) was incubated for 30 min at 37 °C and then placed on ice in the dark until it was required. In order to determine their intracellular ROS scavenging activities, B16F10 cells (1 × 10^4^/well) were seeded in black 96-well plates and incubated for 24 h. The cells were then co-treated with the analogs and SIN-1 (3-morpholinosydnonimine) (10 µM), as previously described [47], for 2 h. DCFH-DA (20 µM) was added for 30 min in the dark and the fluorescence intensities of the oxidized DCFH were measured using a microplate reader (Tecan, Männedorf, Switzerland) at excitation and emission wavelengths of 485 and 530 nm, respectively. Trolox was used as the positive control.

#### 2.2.13. Extraction of Cytosolic and Nuclear Proteins from Cells

B16F10 cells were treated with five concentrations (0, 2.5, 5, 10, and 20 µM) of analog **2** for 24 h, washed twice with cold PBS, and then harvested. Pellets were then suspended in buffer A [NaBu, β-glycerophosphate, 1 M HEPES, 100 mM MgCl, 500 mM DTT, 100 mM NaF, 10 mM Na-orthovanadate, pepstatin, aprotinin, leupeptin, 1% NP-40, and PMSF], incubated on ice for 20 min, and then centrifuged at 12,000 rpm for 10 min at 4 °C. The supernatants that were obtained contained cytosolic fractions. The pellets were resuspended in buffer B [NaBu, β-glycerophosphate, 1 M HEPES, 5 M NaCl, 10 mM Na-orthovanadate, pepstatin, aprotinin, leupeptin, 100 mM NaF, 100 mM EDTA, and PMSF], incubated on ice for 30 min, and then centrifuged at 12,000 rpm for 20 min at 4 °C. The resultant supernatants contained nuclear fractions.

#### 2.2.14. Western Blot Analysis

Protein concentrations were determined using a BCA™ protein assay kit (Pierce, Rockford, IL, USA). Briefly, lysed samples were boiled for 10 min in gel-loading buffer (0.125 M Tris-HCl, pH 6.8, 4% SDS, 10% 2-mercaptoethanol, and 0.2% bromophenol blue) at a volume ratio of 1:1. The total protein equivalents were separated by SDS-PAGE (sodium dodecyl sulfate-polyacrylamide gel electrophoresis) using 9% acrylamide gels and then transferred to polyvinylidene fluoride (PVDF) membranes (Millipore, Burlington, MA, USA) at 25 V for 10 min using a semi-dry transfer system (Bio-Rad Laboratories, Hercules, CA, USA). The membranes were immediately placed in blocking buffer (5% non-fat milk) in TBST (10 mM Tris (pH 7.5), 100 mM NaCl, and 0.1% Tween-20 at room temperature for 1 h. The membranes were then incubated with the appropriate specific primary antibodies at 4 °C overnight and treated with horseradish peroxidase-conjugated anti-mouse or anti-goat antibodies (1:5000) for 1 h at 25 °C. The specific antibodies for tyrosinase, MITF, β-actin, and TFIIB were purchased from Santa Cruz Biotechnology (Santa Cruz, CA, USA). The protein bands were visualized using the SuperSignal^®^ West Pico Chemiluminescent Substrate kit (Advansta, San Jose, CA, USA) and the Davinch-Chemi™ (Davinch-K, Seoul, Korea). The western blot data were quantified using CS Analyser 3.2 (Densitograph) image analysis software (http://www.attokorea.co.kr, accessed on 4 March 2022).

#### 2.2.15. RNA Isolation and Quantitative Real-Time PCR (qRT-PCR)

The total RNAs from the cells were purified using the RiboEx Total RNA solution (GeneAll Biotechnology, Seoul, Korea) and the cDNA was synthesized from the total RNA (2 µg) using SuPrimeScript RT Premix with random hexamer (GeNet Bio, Daejeon, Korea), according to the manufacturers’ instructions. Real-time qPCR was performed using SensiFASTTM SYBR^®^ No-ROX dye (Bioline, London, UK) and a CFX Connect System (Bio-Rad Laboratories, Hercules, CA, USA). The qRT-PCR primers that are specifically for *tyrosinase*, *TRP-1*, *TRP-2*, and *β-actin* were obtained from Bioneer Inc. (Daejeon, Korea). The relative gene expressions were calculated using the standard curve method and β-actin as the internal control. The primer sequences are displayed in Table 1.

#### 2.2.16. Statistical Analysis

A one-way analysis of variance (ANOVA) followed by the Newman–Keuls test was used in order to assess significant differences between the treatments. The analysis was conducted using GraphPad Prism 5 software (La Jolla, CA, USA). The results are the means ± the standard errors of means (SEMs) and two-sided *p*-values of <0.05 were considered to be statistically significant.

## 3. Results and Discussion

### 3.1. Chemistry

The (*Z*)-BPTT analogs that were examined have been previously synthesized from 3-phenyl-2-thioxothiazolidin-4-one (analog **14**) with the appropriate benzaldehydes [48,49,50] or by one-pot reactions between aniline, benzaldehyde, and *C*-(dithiocarboxy)formic acid [51]. Based on considerations of the flexibility and reaction yields, we used 13 (*Z*)-BPTT analogs using the former method (Figure 1). First, analog **14** was synthesized by cyclizing phenyl isothiocyanate with thioglycolic acid in the presence of triethylamine and this was followed by Knoevenagel condensation using different benzaldehydes substituted with a bromo, hydroxyl, methoxy, or ethoxy group on the phenyl ring (Table 2 and Figure 1). The condensation of analog **14** with these benzaldehydes in the presence of NaOAc in acetic acid gave analogs **1**, **2**, and **4**–**13** at yields of 73–90%. However, the reaction between analog **14** and 2,4-dihydroxybenzaldehyde under the same reaction conditions did not afford the desired compound, but a reaction between 2,4-dihydroxybenzaldehyde and analog **14** in the presence of a catalytic amount of 0.3 equiv of piperidine (a weak base) generated analog **3** in a 30% yield. The double bond geometries of the (Z)-BPTT analogs were assigned using ^13^C NMR ^1^H,^13^C coupling constants that were obtained in proton-coupled mode. Nair et al. reported that vicinal ^1^H,^13^C-coupling constants could be used in order to determine the configuration of trisubstituted exocyclic C, C-double bonds [52]. For example, they found that the ^1^H,^13^C-coupling constants of *N*-methyl-*N*,2,3-triphenylacrylamide depended on double bond geometry ((*E*)-isomer: ^3^*J_cis_*
_(*C*(1), *H*-C(3))_ = 6.8 Hz, (*Z*)-isomer: ^3^*J_trans_*
_(*C*(1), *H*-C(3))_ = 11.5 Hz) (Figure 2). Furthermore, the ^3^*J_cis_* values ranged from 3.6 to 7.0 Hz, whereas the ^3^*J_trans_* values were typically >10 Hz. The ^13^C NMR of (*Z*)-BPTT analog **3** was performed in proton-coupled ^13^C mode and the vicinal ^1^H,^13^C-coupling constant (^3^*J*_(*C*(1), *H*-C(3))_) of C1 was 6.6 Hz (Figure 3), which indicated that analog **3** had a (*Z*)-configuration.

### 3.2. The Inhibitory Effects of (Z)-BPTT Analogs on Mushroom Tyrosinase and Calculated Log p Values

The tyrosinase-inhibitory activities were screened against mushroom tyrosinase using L-tyrosine as a substrate, as previously described [40]. Kojic acid, a representative tyrosinase inhibitor, was used as a positive control. The IC_50_ values of the (*Z*)-BPTT analogs were determined using the tyrosinase-inhibitory activities of the test compounds at ≥three different concentrations. All 13 of the (*Z*)-BPTT analogs inhibited tyrosinase activity in a concentration-dependent manner. The IC_50_ values are shown in Table 2, along with the analog substitutions.

Kojic acid had potent tyrosinase-inhibitory activity (IC_50_ = 17.05 ± 3.82 μM). Interestingly, analog **1** (IC_50_ = 1.45 ± 0.04 μM), with a 4-hydroxyl group on the β-phenyl ring of the β-phenyl-α,β-unsaturated carbonyl (PUSC) scaffold, inhibited mushroom tyrosinase 12-fold more than kojic acid. Analogs **12** (IC_50_ = 65.54 ± 9.40 μM) and **13** (IC_50_ = 78.75 ± 8.63 μM) (with a 3-bromo or 3,5-dibromo substitution on the β-phenyl ring of analog **1**, respectively) had significantly lower tyrosinase-inhibitory activities than analog **1**. Similarly, the introduction of an additional 3-methoxyl to the β-phenyl of analog **1** also reduced the tyrosinase-inhibitory activity (analog **4**: IC_50_ = 80.03 ± 3.30 μM). The insertion of an ethoxy group at position 3 of the β-phenyl of analog **1** further reduced the tyrosinase-inhibitory activity (analog **5**: IC_50_ > 200 μM). Analog **4** moderately inhibited mushroom tyrosinase, but exchanging the 4-hydroxyl and 3-methoxyl groups on its β-phenyl ring greatly enhanced its inhibitory activity (analog **6**: IC_50_ = 1.88 ± 0.70 μM) by 43-fold. Notably, the insertion of additional alkoxyl or bromo substituents on the β-phenyl ring of analog **1** diminished its tyrosinase-inhibitory potency, whereas the introduction of an additional hydroxyl substituent at positions 2 or 3 of the β-phenyl ring of analog **1** either retained or enhanced its inhibitory activity (analog **2** with a 3-hydroxyl group: IC_50_ = 1.38 ± 0.11 μM and analog **3** with a 2-hydroxyl group: IC_50_ = 0.59 ± 0.03 μM). Analogs **7** and **10** (IC_50_ values > 200 μM) with no hydroxyl group did not exhibit tyrosinase-inhibitory activity, whereas analogs **8** (IC_50_ = 94.29 ± 7.70 μM) and **9** (IC_50_ = 22.15 ± 4.23 μM), in which all of the hydroxyl groups of analogs **2** and **3** were replaced by methoxyl groups, exhibited weak to moderate activity. Of the 13 synthesized (*Z*)-BPTT analogs, analog **3** most potently inhibited mushroom tyrosinase and was 29-fold more potent than kojic acid. Four analogs, numbers **1**, **2**, **3**, and **6**, inhibited mushroom tyrosinase more than kojic acid.

In order to determine how well the (*Z*)-BPTT analogs are absorbed by skin, their log *p* (partition coefficient) values were examined using ChemDraw Ultra Ver. 12.0. The log *p* values of the 13 analogs are provided in Table 2. Kojic acid had a low log *p* value of –2.45, whereas the (*Z*)-BPTT analogs had log *p* values ranging from 3.16 to 5.20. Analogs **2** and **3** with two hydroxyl groups on the β-phenyl ring of the PUSC scaffold had partition coefficients that were 10^5^-fold greater than that of kojic acid (log *p*: 3.16 vs. –2.45). These results suggest that all 13 of the analogs are better absorbed by skin than kojic acid.

### 3.3. Modes of Action of (Z)-BPTT Analogs ***1***–***3*** and ***6***

We investigated the modes of action of analogs **1**, **2**, **3**, and **6** because they inhibited mushroom tyrosinase more than kojic acid did. Lineweaver–Burk plots were used in order to determine the modes of action of these four analogs (Figure 4). Mushroom tyrosinase inhibition plots were obtained using six different L-DOPA (substrate) concentrations and four different concentrations of each analog. In the Lineweaver–Burk plots, the four lines that were generated by the analogs merged at one point on the y-axis, indicating that the maximum reaction rate (V_max_) of each analog was independent of the analog’s concentration. On the other hand, the Michaelis–Menten constants (K_M_) increased upon the increase of the analogs’ concentrations (Table 3). The K_M_ values of the analogs were 1.042, 1.193, 1.433, and 2.159 mM at analog **1** concentrations of 0, 1, 2, and 4 µM; 1.077, 2.117, 2.899, and 4.865 mM at analog **2** concentrations of 0, 2.5, 5, and 10 µM; 0.793, 1.392, 2.015, and 3.739 mM at analog **3** concentrations of 0, 0.4, 0.8, and 1.6 µM; and 1.030, 1.243, 1.531, and 1.885 mM at analog **6** concentrations of 0, 1, 2, and 4 µM. The mean of the V_max_ values that were determined from the Lineweaver–Burk plots was 0.0171 mM/min. In order to determine the inhibition constants (K_i_) for the mushroom tyrosinase–analog complexes, the Lineweaver–Burk plots were converted into their corresponding Dixon plots (Figure 5), in which 1/V (where V is the initial reaction velocity) was plotted against the inhibitor concentration [I] at different L-DOPA [S] concentrations. The K_i_ values that were obtained for analogs **1**, **2**, **3**, and **6** were 4.67 × 10^−6^, 3.18 × 10^−6^, 7.08 × 10^−7^, and 3.23 × 10^−6^ M, respectively, indicating that analog **3** had the greatest binding affinity for mushroom tyrosinase. Furthermore, these results show that analogs **1**, **2**, **3**, and **6** are competitive inhibitors of mushroom tyrosinase.

### 3.4. In Silico Studies

In silico studies were performed on (*Z*)-BPTT analogs **1**–**3** and **6** using the crystal structure of mushroom tyrosinase and a human tyrosinase (*h*TYR) homology model. Docking studies were conducted using Schrodinger Suite (release 2021-1). As the software did not provide docking results for analog **6**, only the docking results for analogs **1**–**3** are provided in Figure 6 and Figure 7. Kojic acid was used as a reference standard.

#### 3.4.1. Docking Studies on Analogs **1**–**3** and Kojic Acid with Mushroom Tyrosinase

In order to determine the binding affinities of analogs **1**–**3** and kojic acid with mushroom tyrosinase, the crystal structure of mushroom tyrosinase (PDB ID: 2Y9X) was imported from the Protein Data Bank database and docked with analogs **1**–**3** or kojic acid. The binding interactions between these compounds and mushroom tyrosinase are shown in Figure 6 in 2D and 3D formats. According to the results that were obtained, analogs **1**–**3** occupied the same active binding pocket as kojic acid. Analogs **2** (−5.9 kcal/mol) and **3** (−6.0 kcal/mol) had higher binding affinities than kojic acid (−4.5 kcal/mol), whereas analog **1** (−3.3 kcal/mol) had a higher docking score than analog **2**, analog **3**, and kojic acid. Kojic acid and analog **2** interacted with two copper ions. The hydroxyl of kojic acid coordinated with Cu400 and Cu401 and the phenolic 4-OH group on the β-phenyl ring of analog **2** formed salt bridges with Cu400 and Cu401 at distances of 2.19 Å and 2.24 Å, respectively. Also, in the same manner as kojic acid, the phenyl ring of analog **2** interacted with His263 by pi–pi stacking. The phenolic 4-OH group of analog **3** also formed salt bridges with Cu400 and Cu401 at distances of 2.30 Å and 2.34 Å, respectively, and the β-phenyl ring of analog **3** formed two pi–pi interactions with histidine residues His259 and His263. In addition, the *N*-phenyl moiety of analog **3** formed a pi–cation interaction with Arg268. In analog **1**, the β-phenyl ring formed a pi–pi interaction with His259, which led to a lower binding affinity than those of the kojic acid or analogs **2** or **3**.

Comparisons of the binding interactions between mushroom tyrosinase and kojic acid or analogs **1**, **2**, or **3** indicated that the interaction between the 4-OH of the β-phenyl ring of **1**, **2**, or **3** and the two copper ions contributed much to the binding affinity with mushroom tyrosinase. Also, the additional OH on the β-phenyl ring of analogs **2** and **3** at positions 2 or 3 appeared to enhance their binding affinity. Overall, (*Z*)-BPTT analogs **1**, **2**, and **3** appeared to inhibit mushroom tyrosinase directly by acting in the same manner as kojic acid at the active site of the mushroom tyrosinase.

#### 3.4.2. Docking Studies of Interactions between Human Tyrosinase and Analogs **1**–**3** and Kojic Acid

Since no crystal structure was available for human tyrosinase in the PDB database, we first created a homology model of human tyrosinase based on tyrosinase-related protein-1 (TRP-1) so as to estimate the binding affinities between human tyrosinase and analogs **1**–**3** and kojic acid. The binding interactions are shown in 2D and 3D formats in Figure 7. The results showed that analogs **1**–**3** occupied the same active binding pocket as kojic acid. As was observed for mushroom tyrosinase, analogs **2** (−5.7 kcal/mol) and **3** (−6.2 kcal/mol) had lower docking scores than kojic acid (−4.4 kcal/mol) with the active site of the human tyrosinase homology model. Analog **1** (−1.0 kcal/mol) had a much higher docking score than analog **2**, analog **3**, and kojic acid for the binding pocket of the human tyrosinase homology model. In kojic acid, the branched hydroxymethyl group coordinated with zinc (Zn7), the pyranone ring formed a pi–pi stacking interaction with His367, and 3-OH formed a hydrogen bond with Ser375. On the other hand, the 4-hydroxyl group on the β-phenyl ring of analog **2** formed two salt bridges with Zn6 and Zn7 at distances of 2.24 and 2.65 Å, respectively. Like kojic acid, the β-phenyl ring of analog **2** also interacted with His367 by pi–pi stacking and the 3-hydroxyl group on the β-phenyl ring formed a hydrogen bond with Ser380. Analog **3**, like analog **2**, formed two salt bridges with the two zinc ions of the human tyrosinase model at 2.21 Å and 2.32 Å, which were shorter than the salt bridges that were generated by analog **2**. These shorter salt bridge distances may explain why analog **3** had slightly greater binding affinity than analog **2**. In addition, analog **3** formed a pi–pi stacking interaction with His367, which was also used in the same manner by kojic acid and analog **2**. As compared with analogs **2** and **3**, analog **1** showed a reversed conformation in the active site of the human tyrosinase model. The β-phenyl rings of analogs **2** and **3** were located close to the zinc ions; whereas, in analog **1**, the *N*-phenyl ring was located near these ions. Analog **1** only formed a pi–pi stacking interaction with His367 using its *N*-phenyl ring, which indicated that analog **1** has a lower binding affinity for human tyrosinase than analogs **2** or **3** or kojic acid. These results suggest that (*Z*)-BPTT analogs **2** and **3** are potential human tyrosinase inhibitors.

### 3.5. Effects of Analogs ***1***, ***2***, ***3***, and ***6*** on Cell Viability

Since the four (*Z*)-BPTT analogs **1**, **2**, **3**, and **6** exhibited greater mushroom tyrosinase-inhibitory activity than kojic acid, we selected these analogs for further study in cellular experiments. The cell viabilities of analogs **1**–**3** and **6** were evaluated in B16F10 murine melanoma cells. An EZ-Cytox analysis was performed in order to assess the cells’ viabilities. B16F10 cells were treated with six different concentrations (0, 1, 2, 5, 10, and 20 μM) of each analog and incubated for 24 and 48 h in a humidified 5% CO_2_ atmosphere before adding the EZ-Cytox solution. The optical densities of the wells were measured at 450 nm using a microplate reader.

The effects of analogs **1**–**3** and **6** on the B16F10 cells’ viabilities are shown in Figure 8. No significant cytotoxic effect was found after treatment with any analog at ≤20 µM for 24 or 48 h. Therefore, cell-based assays of cellular tyrosinase activity and melanogenesis were performed at concentrations of ≤20 µM in B16F10 cells.

### 3.6. Effects of Analogs ***1***, ***2***, ***3***, and ***6*** on Melanin Production in B16F10 Cells

B16F10 cells were seeded in 6-well culture plates and pretreated with analogs **1**, **3**, and **6** at concentrations of 0, 5, 10, or 20 µM and with analog **2** at 0, 2.5, 5, or 10 µM for 1 h. α-Melanocyte-stimulating hormone (α-MSH; 1 µM) and 3-isobutyl-1-methylxanthine (IBMX; 200 µM) were added sequentially in order to enhance melanin production. After incubation for 48 h at 37 °C, the melanin contents were assessed by measuring the well optical densities at 405 nm using a microplate reader. Kojic acid (10 or 20 µM) was used as a positive control.

The inhibitory effects of analogs **1**–**3** and **6** on melanin production are shown in Figure 9. The B16F10 cells that were co-stimulated with α-MSH and IBMX had melanin contents that were more than 3-fold greater than those of the untreated controls. All four analogs dose-dependently reduced the α-MSH plus IBMX-induced increases in the melanin contents. Analogs **1**, **3**, and **6** at 5 µM were as effective as kojic acid at 20 µM and at 20 µM they reduced the α-MSH plus IBMX-induced increases in the melanin contents by 56.3, 61.3, and 50.9%, respectively, while kojic acid at 20 µM reduced the induced increase in the melanin content by only 22.9%. Analog **2** at 2.5 µM suppressed α-MSH plus IBMX-induced melanogenesis as effectively as kojic acid at 10 µM, and at 10 µM decreased α-MSH plus IBMX-induced increase in melanin content by 89.6%, which was 4.1-fold higher than the suppression that was yielded from kojic acid at 10 µM, which reduced the α-MSH plus IBMX-induced increase in melanin content by 21.8%. Notably, although analog **3** inhibited mushroom tyrosinase activity slightly more than analog **2**, analog **2** inhibited melanogenesis much more than analog **3**. These results suggest that analog **2** is a promising inhibitor of melanogenesis in mammalian cells.

### 3.7. Inhibitory Effects of Analogs ***1***, ***2***, ***3***, and ***6*** on Cellular Tyrosinase Activities in B16F10 Cells

In order to determine whether the inhibitory effects of the (*Z*)-BPTT analogs on melanin production were due to the inhibition of cellular tyrosinase activity, we also examined the inhibitory effects of analogs **1**, **2**, **3**, and **6** on cellular tyrosinase activity in B16F10 melanoma cells. Briefly, B16F10 cells were seeded in 6-well culture plates and pretreated with four different concentrations (0, 5, 10, or 20 µM) of each of the four analogs for 1 h, and then the cells were co-stimulated with α-MSH and IBMX for 48 h at 37 °C so as to increase their cellular tyrosinase activity. Their optical densities (ODs) were then measured using a microplate reader at 492 nm in order to quantify the dopachrome production. The inhibitory effect of each analog on cellular tyrosinase activity was evaluated using these ODs. Kojic acid (20 µM) was used as a positive control.

The measured tyrosinase-inhibitory activities are shown in Figure 10. Co-stimulation with α-MSH and IBMX enhanced cellular tyrosinase activity as compared to the activity level of the non-treated controls (100%). Tyrosinase activities were concentration-dependently reduced by the analog pretreatments. Specifically, analogs **1**, **3**, and **6** at 20 µM inhibited α-MSH plus IBMX-induced cellular tyrosinase activity more than kojic acid at 20 µM. Analog **2** had the greatest effect and at 5 µM significantly inhibited tyrosinase activity more than kojic acid did at 20 µM. In addition, analog **2** at 10 µM greatly decreased the α-MSH plus IBMX-induced the tyrosinase activity, which was lower than the tyrosinase activity level of the control (100%). Notably, analog **3** inhibited mushroom tyrosinase-inhibitory activity slightly more than analog **2**, while analog **2** inhibited the activity of B16F10 tyrosinase much more than analog **3**. The tyrosinase inhibition pattern was similar to the melanin content reduction pattern that is shown in Figure 9, suggesting that the anti-melanogenesis effects of (*Z*)-BPTT analogs **1**–**3** and **6** are mainly due to their tyrosinase-inhibitory activities.

### 3.8. DPPH Radical Scavenging Activities

The DPPH (2,2-diphenyl-1-picrylhydrazyl) radical scavenging abilities of the (*Z*)-BPTT analogs were investigated using l-ascorbic acid (vitamin C) as a positive control. The final concentrations of the analogs and of the positive control in the test samples and in the DPPH methanol solution were 1 and 0.18 mM, respectively. The scavenging activities of the analogs were assessed by measuring their optical densities at 517 nm after standing the samples in the dark for 30 min. Four (*Z*)-BPTT analogs, analogs **2**–**5,** exhibited moderate to strong DPPH radical scavenging activities as compared with l-ascorbic acid (Figure 11). Analog **1,** with a 4-hydroxyl group on the β-phenyl ring, showed weak DPPH radical scavenging activity (14.7%). However, the insertion of an additional 3-alkoxyl group on the β-phenyl ring greatly increased the radical scavenging activity to 78.7% for analog **4** and 77.5% for analog **5**. The radical scavenging activity was slightly improved when additional methoxyl groups were introduced at positions 3 and 5 of the β-phenyl ring of analog **1**. On the other hand, bromo groups at position 3 or positions 3 and 5 on the β-phenyl ring of analog **1** did not influence DPPH radical scavenging activity. Of the 13 (*Z*)-BPTT analogs, analog **2** with a catechol (3,4-dihydroxyphenyl) moiety exhibited the strongest DPPH radical scavenging activity at 92.2%, comparable to L-ascorbic acid’s 96.9% scavenging activity. In addition, analog **3** with a resorcinol (2,4-dihydroxyphenyl) moiety exerted potent radical scavenging activity (57.0% inhibition). Analogs **7**–**10** with no hydroxyl group were found to have no or weak DPPH radical scavenging effect. Notably, analog **4** with a 4-hydroxy-3-methoxyl group on the β-phenyl ring exhibited strong radical scavenging activity, whereas analog **6**, in which the positions of the 4-hydroxyl group and the 3-methoxyl group of analog **4** were exchanged, had poor radical scavenging activity (27.2%). These results suggest that phenolic hydroxyl groups and their positions on the β-phenyl ring importantly influence DPPH radical scavenging activity.

### 3.9. Effects of the 13 Analogs on ABTS Free Radical Scavenging Activity

The effects of analogs **1**–**13** on ABTS (2,2′-azino-bis(3-ethylbenzothiazoline-6-sulfonic acid)) free radical scavenging activity were investigated using trolox as a positive reference control. The concentrations of the analogs and trolox in the test samples were 100 µM. The free radical scavenging activities were assessed by measuring the optical densities at 734 nm after standing in the dark for 2 min. The ABTS free radical scavenging activities of the analogs are provided in Figure 12. Notably, the ABTS free radical scavenging results of the (*Z*)-BPTT analogs were similar to the DPPH radical scavenging results. Five analogs, **2**–**5** and **11**, exhibited potent radical scavenging activities as compared with the other analogs, as was observed for the DPPH radical scavenging activities. Analogs **4**, **5**, and **11** exhibited moderate ABTS scavenging activities at 61.5%, 51.2%, and 57.2%, respectively. Analog **3,** with a resorcinol moiety, had moderate DPPH radical scavenging ability but exhibited strong ABTS radical scavenging activity (82.0% inhibition). As was observed for the DPPH radical scavenging activity, analog **2** (96.9% inhibition), with a catechol moiety, exhibited the strongest ABTS free radical scavenging activity and was comparable to trolox (99.8% inhibition) in this respect. Analogs **7**–**10**, with no hydroxyl groups on the β-phenyl ring, had weak ABTS and DPPH radical scavenging activities. Analog **1,** with a 4-hydroxyl group on the β-phenyl ring, also had weak ABTS free radical scavenging activity, but the insertion of an additional 3,5-dimethoxyl group on the β-phenyl ring greatly increased its radical scavenging activity (57.2% for analog **11**). However, the introduction of a bromine into positions 3 and 5 of the β-phenyl ring of analog **1** slightly decreased its ABTS free radical scavenging activity (analog **13**) as compared with analog **1**. However, the introduction of one bromo substituent at position 3 of analog **1** did not influence the ABTS free radical scavenging activity (analog **12**).

### 3.10. Effects of the Analogs on Intracellular ROS Scavenging Activity

We investigated the effects of the analogs on intracellular ROS (reactive oxygen species) because, reportedly, ROS might regulate melanogenesis in melanoma cells [53]. The principle of the intracellular ROS assay that was used is that DCFH-DA (2′,7′-dichlorodihydrofluorescein diacetate) spreads across the cell membrane and is enzymatically hydrolyzed by esterases to DCFH (2′,7′-dichlorodihydrofluorescein), which reacts with ROS in order to generate DCF (2′,7′-dichlorofluorescein), which is fluorescent [46]. Thus, the intracellular ROS scavenging activities of (*Z*)-BPTT analogs **1**–**3** and **6** were investigated in vitro using DCFH-DA. SIN-1 (3-morpholinosydnonimine) and trolox were used as a RNS/ROS generator [54] and a positive control, respectively. Treatment with 10 µM of SIN-1 greatly enhanced intracellular ROS generation, as compared to the level of generation that is seen in untreated B16F10 cells (Figure 13). Analogs **1** and **6** did not show ROS scavenging activity at concentrations ≤ 20 µM, but analog **3**, which possessed a resorcinol moiety, at 20 µM had a ROS scavenging effect that was similar to that of trolox. On the other hand, analog **2,** with a catechol moiety, significantly and concentration-dependently reduced SIN-1-induced increases in the intracellular ROS activity and it did so to a greater extent than trolox. These results suggested that (*Z*)-BPTT analogs, especially analog **2**, effectively reduced SIN-1-induced increases in intracellular ROS levels. A comparison of the structures and ROS scavenging activities of the (*Z*)-BPTT analogs revealed that the catechol moiety (the 3,4-dihydroxyphenyl group) markedly influenced intracellular ROS scavenging activity, a finding which concurs with the findings of previous reports [55,56,57]. These results suggest that the anti-melanogenesis effect that is created by analog **2** is probably, at least in part, due to the melanogenesis inhibition that is caused by intracellular ROS scavenging.

### 3.11. Effect of (Z)-BPTT Analog ***2*** on the Expressions of Melanogenesis-Associated Genes in B16F10 Cells

Because analog **2** inhibited melanogenesis in B16F10 cells more than analogs **1**, **3**, and **6**, we investigated the mechanism that is responsible for its anti-melanogenic effect. Western blotting was used to determine the cytosolic tyrosinase protein levels and nuclear MITF levels in total cell lysates (Figure 14A,B). The results revealed that analog **2** treatment significantly and concentration-dependently reduced the tyrosinase level. However, contrary to the reduction of tyrosinase expression at low concentrations (2.5–10 uM), MITF expression instead increased at the same concentrations. Due to this discrepancy, we examined the effect of analog **2** on MITF target gene expressions such as *tyrosinase*, *TRP-1*, and *TRP-2*, which are required for melanin biosynthesis [58,59]. The mRNA levels were determined by quantitative RT-PCR (qRT-PCR) by comparing the mRNA expression levels in analog **2**-treated B16F10 cells with those of non-treated controls. The expression levels of all three of the enzymes were significantly and concentration-dependently reduced by analog **2**, which suggests that analog **2** downregulated MITF expression and, thus, suppressed the expressions of *tyrosinase*, *TRP-1*, and *TRP-2* (Figure 14C). These results suggest that the depigmenting effect of analog **2** is in part due to its downregulation of melanogenic enzymes.

## 4. Conclusions

Thirteen (*Z*)-BPTT analogs were synthesized based on the structural characteristics of two potent tyrosinase inhibitors. The analogs’ structures were determined by ^1^H and ^13^C NMR and low- and high-resolution mass spectroscopy. In particular, the double bond geometries were elucidated using ^1^H,^13^C-coupling constants. The 13 analogs were evaluated for mushroom tyrosinase-inhibitory activity and four analogs (analogs **1**–**3** and **6**) were found to be more potent than kojic acid. Kinetic studies performed using Lineweaver–Burk plots indicated that all four of these analogs were competitive mushroom tyrosinase inhibitors and this was supported by docking results. A human tyrosinase homology model was prepared and docked with analogs **1**–**3** and analogs **2** and **3** were found to have greater binding affinity than kojic acid. Four analogs, **1**–**3** and **6,** dose-dependently inhibited cellular tyrosinase activity and melanin production in B16F10 cells more potently than kojic acid did. In particular, analog **2**, which had a catechol moiety, was much more potent than the other three analogs and kojic acid. In addition, five analogs showed moderate to strong radical scavenging activities against DPPH and ABTS and analog **2** showed the greatest radical scavenging activity, which was comparable to those of the positive controls, l-ascorbic acid and trolox. Furthermore, analog **2** more potently scavenged ROS than trolox and it significantly and concentration-dependently decreased tyrosinase and MITF protein levels and the expressions of melanogenesis-related genes viz. *tyrosinase* and *TRP-1* and *-2*. These results have demonstrated that analog **2** has anti-melanogenic properties that are derived from dual modes of action, namely, (i) by directly inhibiting tyrosinase activity and (ii) by suppressing the expressions of melanogenesis-related proteins and genes.

## Data Availability

The data are contained within the article and Appendix A.

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
