# Peer review of "Identification of a Novel Class of Anti-Melanogenic Compounds, (Z)-5-(Substituted benzylidene)-3-phenyl-2-thioxothiazolidin-4-one Derivatives, and Their Reactive Oxygen Species Scavenging Activities"

_antioxidants, 2022, doi:10.3390/antiox11050948_

Round 1

Reviewer 1 Report

The authors created 13 new (Z)-BPTT analogs using known tyrosinase inhibitors 5-HMT and Compound 1 as lead compounds and investigated their effects on melanogenesis.
They investigated their ability to scavenge ROS and inhibit TYRs activity and expression, and Analog 2 was found to have a more substantial inhibitory effect on melanogenesis than kojic acid.
This is an intriguing study that demonstrates the utility of BPTT analogs, and this reviewer believes it should be published in the Antioxidants. However, before publication, this reviewer requests the following additions and corrections.

Minors:

1) MITF is the critical regulator for TYR expression; however, in Figure 14, at low concentrations of the analog 2 (5-10 uM), TYR expression is reduced regardless of MITF expression level, which should be explained and discussed in the Results and Discussion part.

2) Regarding the WB band of TYR in the original image file, the TYR is detected initially as a band larger than 70 kDa, but the authors' data shows the TYR size as approximately 62 kDa. Therefore, some explanation for this is required.

Typos:

3) Line 313,
Streptomycin/penicillin (100 IU/100 μg/mL) should be 
Streptomycin/penicillin (100 μg/mL/100 IU/mL).

4) Line 370,
ATBS should be ABTS.

Author Response

Reviewer 1.

The authors created 13 new (Z)-BPTT analogs using known tyrosinase inhibitors 5-HMT and Compound 1 as lead compounds and investigated their effects on melanogenesis.
They investigated their ability to scavenge ROS and inhibit TYRs activity and expression, and Analog 2 was found to have a more substantial inhibitory effect on melanogenesis than kojic acid.
This is an intriguing study that demonstrates the utility of BPTT analogs, and this reviewer believes it should be published in the Antioxidants. However, before publication, this reviewer requests the following additions and corrections.

Minors:

1) MITF is the critical regulator for TYR expression; however, in Figure 14, at low concentrations of the analog 2 (5-10 uM), TYR expression is reduced regardless of MITF expression level, which should be explained and discussed in the Results and Discussion part.

Answer: Thank you for your valuable comment. As the reviewer suggested, we have discussed it in the Results and Discussion part. Please see the corresponding text body.

2) Regarding the WB band of TYR in the original image file, the TYR is detected initially as a band larger than 70 kDa, but the authors' data shows the TYR size as approximately 62 kDa. Therefore, some explanation for this is required.

Answer: Thank you for your valuable comment. Unfortunately, contrary to what the reviewer and we expected, there is no WB band in the 70 kDa range. So, we searched a lot of related material. As in your comment, many papers report that the WB band of TYR appears at about 70 kDa, but some papers report that the band appears at about 60 kDa. Please see the following articles: (1) Research in Pharmaceutical Sciences, December 2018; 13(6): 533-545 (2) Natural Product Communications, June 2019: 1–6.

Typos:

3) Line 313,
Streptomycin/penicillin (100 IU/100 μg/mL) should be 
Streptomycin/penicillin (100 μg/mL/100 IU/mL).

Answer: Thank you for your kind indication. We have revised them.

4) Line 370,
ATBS should be ABTS.

Answer: Thank you for your kind indication. We have revised it.

Reviewer 2 Report

The article entitled “Identification of a novel class of anti-melanogenic compounds, 2 (Z)-5-(substituted benzyli- 3 dene)-3-phenyl-2-thioxothiazolidin-4-one derivatives and reactive oxygen species scavenging activities” describes a synthesis of a series of novel ((Z)-BPTT) analogs and their potential to inhibit melanogenesis at different stages and with different mechanism (tyrosinase inhibition, melanoma cell viability suppression, radical scavenging potential). The work is valuable as it fits into a trend of searching new anti-melanogenic agents that could be used in the treatment not only of hyperpigmentation but also melanoma which is a severe disease with high death rate. The authors made effort to investigate biological activities of the newly synthesized agents, however there are some questions and suggestions:

  1. Why did the Authors use different substrates for tyrosinase in tyrosinase inhibition assay and kinetic studies of tyrosinase (L-tyrosine and L-DOPA, respectively)? Those substrates are related to different tyrosinase activities (monophenolase or diphenolase).
  2. In DPPH assay inline 361 are mentioned positive controls – distilled water is not a positive control as it does not have radical scavenging properties.
  3. The safety of the most promising compounds could be also assessed and its activity towards normal keratinocytes could be investigated to check the potential of therapeutic use.

Author Response

Reviewer 2.

The article entitled “Identification of a novel class of anti-melanogenic compounds, 2 (Z)-5-(substituted benzyli- 3 dene)-3-phenyl-2-thioxothiazolidin-4-one derivatives and reactive oxygen species scavenging activities” describes a synthesis of a series of novel ((Z)-BPTT) analogs and their potential to inhibit melanogenesis at different stages and with different mechanism (tyrosinase inhibition, melanoma cell viability suppression, radical scavenging potential). The work is valuable as it fits into a trend of searching new anti-melanogenic agents that could be used in the treatment not only of hyperpigmentation but also melanoma which is a severe disease with high death rate. The authors made effort to investigate biological activities of the newly synthesized agents, however there are some questions and suggestions:

  1. Why did the Authors use different substrates for tyrosinase in tyrosinase inhibition assay and kinetic studies of tyrosinase (L-tyrosine and L-DOPA, respectively)? Those substrates are related to different tyrosinase activities (monophenolase or diphenolase).

Answer: Thank you for your valuable comments. Tyrosinase has both activities as monophenolase and diphenolase. Since we used a substrate for monophenolase in the tyrosinase inhibition assay, it would be better to use the same substrate in the kinetic study of tyrosinase. But we were not aware of it. Your valuable comments will be remembered for future research.

2. In DPPH assay inline 361 are mentioned positive controls – distilled water is not a positive control as it does not have radical scavenging properties.

Answer: Thank you for your kind comments.

We have revised the phrase as follows:

(10 mM l-ascorbic acid in distilled water)

3. The safety of the most promising compounds could be also assessed and its activity towards normal keratinocytes could be investigated to check the potential of therapeutic use.

Answer: Thank you for your valuable comments. We have a plan to do a further research associated with experiments suggested by the reviewer. Thank you for your constructive opinions.